# Physiological Mechanism of EBR for Grain-Filling and Yield Formation of Tartary Buckwheat

**DOI:** 10.3390/plants13233336

**Published:** 2024-11-28

**Authors:** Han Liu, Qiang Wang, Ting Cheng, Yan Wan, Wei Wei, Xueling Ye, Changying Liu, Wenjun Sun, Yu Fan, Liang Zou, Laichun Guo, Dabing Xiang

**Affiliations:** 1Key Laboratory of Coarse Cereal Processing, Ministry of Agriculture and Rural Affairs, Sichuan Engineering & Technology Research Center of Coarse Cereal Industrialization, School of Food and Biological Engineering, Chengdu University, Chengdu 610106, China; liuhan1003@stu.cdu.edu.cn (H.L.); wangqiangeternity@icloud.com (Q.W.); chengting@stu.cdu.edu.cn (T.C.); yanwan@cdu.edu.cn (Y.W.); shirlingye@gmail.com (X.Y.); liuchangying@cdu.edu.cn (C.L.); sunwenjun@cdu.edu.cn (W.S.); fandavi@163.com (Y.F.); zouliang@cdu.edu.cn (L.Z.); 2Sichuan Engineering & Technology Research Center of Coarse Cereal Industrialization, Chengdu University, Chengdu 610106, China; 3Agronomy College, Jilin Agricultural University, Changchun 130118, China; 4Baicheng Academy of Agricultural Sciences, No. 17, Sanhe Road, Taobei District, Baicheng 137000, China

**Keywords:** Tartary buckwheat, grain-filling, EBR, yield, physiology and biochemistry

## Abstract

Tartary buckwheat is characterized by its numerous inflorescences; however, the uneven distribution of resources can lead to an overload in certain areas, significantly limiting plant productivity. Plant growth regulators effectively modulate plant growth and development. This study investigated the effects of three concentrations of brassinosteroids (EBR) on the Tartary buckwheat cultivar with high seed-setting rates, specifically Chuanqiao No. 1 (CQ1), and low seed-setting rates, namely Xiqiao No. 1 (XQ1), through field experiments. The goal was to investigate how EBR regulates buckwheat grain-filling, enhancing the seed-setting rates, and to understand the physiological mechanisms behind this improvement. The results indicated that EBR treatment followed the typical “S” type growth curve of crops, resulting in an increase in the Tartary buckwheat grain-filling rate. Varieties with high seed-setting rates demonstrated a greater capacity for grain-filling. EBR was observed to regulate hormone content, enhance the photosynthetic capacity of Tartary buckwheat, and increase yield. This was accomplished by enhancing the accumulation of photosynthetic products during the grain-filling period. Specifically, EBR elevated the activity of several key enzymes, including pre-leaf sucrose phosphate synthase (SPS), seed sucrose synthase (SS), late grain-filling acid invertase (AI), grain-filling leaf SPS, and grain SS. These changes led to an increased accumulation of sucrose and starch from photosynthetic products. In summary, the G2 concentration of EBR (0.1 mg/L) demonstrated the most significant impact on the seed-setting rates and yield enhancement of Tartary buckwheat.

## 1. Introduction

Tartary buckwheat (*Fagopyrum tataricum* L.) Gaertn. is an annual herb belonging to the genus Buckwheat in the family Polygonaceae [1]. The seeds of Tartary buckwheat are known for their richness in flavonoids, which have shown promising preventive effects against the high incidence of chronic diseases in humans [2]. Furthermore, compounds like quercetin have been found to inhibit the growth of cancer cells through their antioxidant effects [3], highlighting the potential of Tartary buckwheat in the fields of food and medicine [4].

Tartary buckwheat, characterized by an indeterminate inflorescence [5], exhibits simultaneous flowering and grain-filling. The high ATP consumption during flowering leads to a differentiation between strong and weak grain-filling, resulting in an uneven distribution of materials and an overload of the source. Grain-filling is influenced by both sink strength (in terms of size and activity) and source strength (the ability to provide photoassimilates) [6]. Enhancing source activity or sink strength may increase yield [7]. Carbon is fixed from carbon dioxide into carbohydrates within the chloroplasts of leaf tissues, subsequently accumulating in the cytoplasmic solutes of the same cells. The energy requirements of sink tissues, such as roots, flowers, and seeds, drive the export of sugars from the leaves, primarily in the form of sucrose, through long-distance transport via the phloem, the plant’s vascular tissue [8,9]. This suggests that sucrose source-to-sink transport is a major factor influencing plant growth. Key carbohydrate metabolizing enzymes like SPS and AGPase play a critical role in regulating the metabolic state of source leaves by balancing source–sink dynamics [10]. Sink strength plays a crucial role in controlling plant growth and the photosynthetic rate of source leaves [11]. Therefore, maintaining a balance in the distribution of photosynthetic assimilates from the source to the carbon sink is crucial for successful Tartary buckwheat seed grain-filling [12].

Exogenous phytohormone treatment is an environmentally friendly and efficient method [13], with physiological functions such as regulating crop growth and development, affecting cell division, growth traits, metabolic processes, flowering, seed-setting, and seed development [14]. Crop source activities, including photosynthesis, are influenced by the distribution of photosynthetic assimilates, as well as the regulation of phytohormone metabolism [15]. For example, abscisic acid (ABA) favors the regulation of leaf stomatal closure [16]. Zeatin (ZT) affects photosynthetic function [17]. Indole-3-acetic acid (IAA) regulates leaf development and stomatal opening [18]. Gibberellic acid (GA) increases relative chlorophyll content by maintaining leaf membrane permeability [19]. B9 increases leaf dry matter in sagebrush [20], and ethylene glycol promotes chlorophyll synthesis [21]. Among them, EBR, recognized as one of the top six most important plant hormones globally [22], plays a significant role in regulating plant flowering time, phenotypic composition, and yield formation, as well as in improving stress tolerance [23]. Previous studies found that EBR also significantly increased leaf area in Arabidopsis [24], improved rice grain-filling efficiency [25], and increased wheat grain yield [26], among other important roles. In other crop production, certain concentrations of GA_3_, ethephon, B9, brassinolide, and TIBA can promote plant flowering and fruiting. EBR participates in many physiological functions of plants, but research on its effect on Tartary buckwheat flowering and fruiting, grain-filling, etc., has not yet been reported.

Preliminary experimental research showed that EBR treatment is most effective for promoting Tartary buckwheat growth compared to other plant growth regulators. This study aimed to conduct field trials with Tartary buckwheat varieties that have different seed-setting rates, i.e., a high seed-setting rate (CQ1) and a low seed-setting rate (XQ1). The trials were designed to investigate how EBR impacts Tartary buckwheat growth, development, grain-filling, and yield. The study also examined Tartary buckwheat yield components, flowering and grain-filling dynamics, seed germination characteristics, photosynthetic physiological traits, and the patterns of material accumulation and transportation. This research elucidated the physiological mechanism by which EBR regulates Tartary buckwheat grain-filling and yield, offering a theoretical foundation and technical support for enhancing Tartary buckwheat yield through chemical control and addressing low seed-setting rates or yields in Tartary buckwheat cultivation.

## 2. Results

### 2.1. Effects of Different Growth Regulators on Tartary Buckwheat Seed-Setting and Yield

Tartary buckwheat agronomic properties and yield composition vary under the influence of different plant growth regulators. The GA_3_ treatment significantly increased plant height by 10.19% compared to that of the control (CK), while B9 and ETH treatments decreased height by 9.32% and 3.28%, respectively, compared to that of CK. Among the various treatments, EBR demonstrated notable advantages in enhancing yield, including the total flower number (Figure 1H), seed-setting rate (Figure 1L), number of grains per plant (Figure 1D), grain weight per plant (Figure 1F), mature grain number (Figure 1K), and overall yield (Figure 1G), which increased by 21.80%, 3.51%, 11.58%, 34.88%, 39.12%, and 15.75%, respectively, compared to the results for CK. The EBR treatment resulted in an increase in the number of dead seeds (Figure 1J), as EBR enhanced the setting-rate of Tartary buckwheat. As the total number of flowers increased, there was a corresponding rise in the number of dead seeds. Consequently, our next step will be to elucidate the physiological mechanisms underlying the yield increase attributed to EBR through field experiments.

### 2.2. Effects of EBR on Tartary Buckwheat Grain-Filling and Yield

#### 2.2.1. Effect on Tartary Buckwheat Seed-Setting and Yield

As illustrated in Figure 2A–I, the number of grains per plant of CQ1 and XQ1 increased by 39.47% and 21.7%, respectively, under G2 treatment. The effect of EBR on enhancing the grain weight of Tartary buckwheat plants was most pronounced in G2, with CQ1 and XQ1 experiencing increases of 28.82% and 30.28%, respectively. The yield of Tartary buckwheat exhibited a low-promoting and high-inhibiting effect with increasing EBR concentration. Under G2 treatment, the yields of CQ1 and XQ1 rose by 10.4% and 4.8%, respectively. The solid grain–leaf ratio of CQ1 increased following EBR treatment. The effect of each CQ1 treatment group on yield was greater than that of the source, suggesting that CQ1 is a crop that enhances storage and increases yield. At a concentration of 0.1 mg/L, CQ1 appears to align more closely with the source–sink interaction model. At low concentrations, XQ1 functions as a type that enhances reservoir capacity and increases production. As EBR concentration rises, the influence of the source on output becomes more pronounced than that of the reservoir, favoring the type that increases the source and boosts production. Both low and high concentrations can result in the reservoir exerting a significantly greater influence on yield than the source, reflecting a type that enhances reservoir capacity and production. Conversely, at the G2 concentration, the effect is characterized by an increase in the source, leading to enhanced production. The total grain-to-leaf ratio of XQ1 decreased, indicating that XQ1 is associated with increased source capacity and yield. However, higher and lower concentrations result in the Tartary buckwheat reservoir having less impact on yield than that of the source.

Figure 2J,K illustrates that the seed-setting rates of CQ1 and XQ1 increased by 6.31% and 3.27%, respectively, under G2 treatment. Notably, the grain abortion rate for CQ1 decreased significantly with increasing EBR concentration. In contrast, EBR reduced the grain abortion rate of XQ1 by 1.45% at the G2 concentration, while it increased the rate by 0.21% at the G3 concentration.

#### 2.2.2. The Effect of EBR Treatment on the Grain-Filling Rate of Tartary Buckwheat

As illustrated in Table 1, it is evident that R² exceeds 0.990. This indicates that the Richards equation provides a more accurate simulation of the grain-filling development process following EBR treatment, thereby establishing a reliable basis for assessment. Under G2 treatment, the CQ1 and XQ1 A values increased by 5.26% and 4.77%, respectively, compared to those of the control group. EBR enhances the initial grouting potential of CQ1, with the R0 values for XQ1 and CQ1 being the highest under the G2 treatment. However, as the concentration increases, both A and R0 exhibit a downward trend. Additionally, under G2 treatment, the V_max_ values for CQ1 and XQ1 increased by 15% and 27%, respectively, demonstrating a more pronounced promoting effect of EBR on XQ1. T_max_, which denotes the time required to reach the maximum grain-filling rate, was calculated to be 19.5 days for CQ1 and 22.2 days for XQ1 in the control group. EBR treatment reduced T_max_, with the most significant reduction observed at the G2 concentration; specifically, CQ1 and XQ1 experienced reductions of 1.97 days and 1.44 days, respectively. The average filling rate of Tartary buckwheat grains, represented by $\bar{V}$, follows a similar trend to that of V_max._

### 2.3. Physiological Mechanism of EBR in Regulating Tartary Buckwheat Grain-Filling

#### 2.3.1. Effect on Photosynthetic Parameters

Table 2 demonstrates that a G2 treatment with EBR lasting 10 to 20 days significantly enhances the net photosynthetic rate of Tartary buckwheat during the grain-filling period. This effect is particularly pronounced in CQ1, while the high-concentration EBR G3 treatment leads to a reduction in the net photosynthetic rate. Throughout the entire grain-filling period, the transpiration rate of CQ1 consistently exceeded that of XQ1, exhibiting an overall downward trend, with the most substantial improvement observed in the G2 treatment group. The intercellular CO_2_ concentration (Ci) initially decreased and subsequently increased, reaching its lowest point at 20 days; notably, the Ci concentration of CQ1 remained higher than that of XQ1. The initial fluorescence of Tartary buckwheat during the grain-filling period demonstrated a pattern of first increasing and then decreasing, with CQ1 values generally surpassing those of XQ1, while most treatment groups recorded lower values than those for G0.

As illustrated in Figure 3A–F, the concentrations of chlorophyll a, chlorophyll b, and total chlorophyll exhibited an initial increase followed by a decrease from 10 to 30 days post-flowering, which were all significantly higher than those observed for G0. Thus, a 0.1 mg/L concentration of EBR can enhance the actual photosynthetic capacity of Tartary buckwheat during the grain-filling period, thereby improving light utilization efficiency and promoting photosynthesis.

#### 2.3.2. Effects on Dry Matter Conversion and Enzyme Activity

Figure 4A–C illustrates that during the flowering stage of Tartary buckwheat, the stem dry weight of CQ1 decreased significantly at 30 days. CQ1 allocates more resources to its roots and flowers compared to XQ1, which directs a greater proportion of resources to its stems and leaves. Notably, CQ1 exhibits more leaves during the grain-filling stage than in the flowering stage, indicating a shift in the center of gravity of material transport toward the grain. The intensity of grain-filling in CQ1 surpasses that of XQ1. Conversely, during the grain-filling stage, XQ1 prioritizes material allocation towards developing flowers and stems over that of CQ1. Additionally, EBR treatment enhances the reproductive growth and development of Tartary buckwheat during the grain-filling period, significantly increasing the proportion of grains in both CQ1 and XQ1 under G2 treatment. Compared to the control group, the treatment group demonstrates an increase in the dry matter weight of the entire plant and promotes a higher grain dry matter distribution ratio.

As illustrated in Figure 4D–O, the treatment of EBR at various concentrations significantly enhances the chemical activity of SPS. The SPS activity in the functional leaves during the grain-filling stage peaks at 20 days after flowering, with G2 exhibiting the highest activity, and EBR demonstrating a more pronounced effect on CQ1. The sucrose content exhibits a downward trend; notably, the sucrose levels in the 10d–20d treatment group surpass those of the control group. While SPS enzyme activity increases, the rate of sucrose synthesis in the leaves exceeds the transport rate. Consequently, EBR treatment promotes sucrose synthesis in the leaves during the early 10d–20d period of Tartary buckwheat grain-filling. Conversely, between 20d and 30d, the sucrose content falls below that of the control group, accompanied by a decrease in SPS enzyme activity. This indicates that the sucrose transport rate in the leaves surpasses the synthesis rate. Thus, EBR enhances the sucrose transport rate in the later stages of Tartary buckwheat grain-filling. SS exhibits an upward trend, reaching its maximum value at 20d, while AI shows a fluctuating trend. Each treatment significantly elevates the enzyme activities of SS and AI, with peak activity observed during G2 treatment. The sucrose content in Tartary buckwheat grains initially increases before declining, while starch content trends upward. All treatment groups demonstrated higher levels than those of the control group. EBR significantly increases the sucrose and starch content in Tartary buckwheat grains during the early to middle stages of grain-filling, with the G2 treatment identified as the most effective concentration.

#### 2.3.3. Effects on Endogenous Hormones

Figure 5A–D illustrates that there are no significant changes in IAA and TZ. ABA exhibited a decreasing-increasing-descending trend with increasing EBR concentration in CQ1, with G2 significantly higher than G0. In XQ1, ABA showed an upward trend; however, the overall content in the treatment group remained lower than that of the control group. No significant difference in GA_3_ was observed in CQ1. In XQ1, GA_3_ demonstrated a trend of decreasing and then increasing with the rise in EBR concentration, with the treatment group again showing lower levels than the control group.

### 2.4. Principal Component Analysis

To evaluate the contributions and correlations among various traits of Tartary buckwheat plants at different growth stages, we conducted principal component analysis (Figure 6). The results indicated that EBR treatment primarily enhances the photosynthetic capacity in the early stages, influences dry matter accumulation by promoting enzyme activity in the later stages, and ultimately increases yield. Principal component analysis (PCA) is a multivariate statistical method that simplifies and reduces the dimensionality of high-dimensional data while preserving the original information to the greatest extent possible. We analyzed the contribution rates of Tartary buckwheat-related traits under varying concentrations of growth agents using PCA. The PCA for CQ1 at 20 days demonstrated a contribution rate of 95.8%, with PC1 and PC2 accounting for 80.0% and 15.8%, respectively. In contrast, the PCA for XQ1 at 30 days yielded a contribution rate of 95.7%, with PC1 and PC2 comprising 74.1% and 21.6%, respectively. The vector lengths on the biplot indicate the variability of the indicators across different treatments. Furthermore, we observed that the photosynthetic parameters of various varieties clustered distinctly at specific locations across different time points. According to the projection of variables on PC1, the photosynthetic parameters and starch–sucrose levels are key factors influencing plant biomass. A greater distance between samples in the PCA plot indicates a more significant difference. The figure illustrates that the four treatment groups are highly dispersed in the PCA plot, suggesting substantial differences among them. Notably, the distance between G2 and G0 is the longest, indicating that the G2 treatment enhances crop-filling and yield by improving photosynthetic capacity.

## 3. Discussion

### 3.1. Effects of EBR on the Rate of Tartary Buckwheat Grain-Filling and the Accumulation and Transportation of Substances

The role of EBR in crop grain development [27] and substance accumulation [28] needed further improvement. Previous studies have shown that foliar application of EBR at the end of mung bean vegetative growth will increase seed number and grain yield. This is because the plant’s physiological activation is increased, providing a sufficient nutrient supply for flower growth, a more rapid flowering stage, and better seed development, ultimately leading to higher grain yields [29]. By fitting the grain-filling Richards growth equation, the grain-filling process after EBR treatment conforms to the crop S-shaped growth curve [30] and increases the Tartary buckwheat grain weight. EBR significantly increased the R0 of Tartary buckwheat, while the yield and seed-setting rate of XQ1 under the G2 treatment were comparable to those of the CQ1 blank group, indicating that varietal differences in seed-setting rate and yield could be compensated by chemical regulation. EBR also increased V_max_ and advanced T_max_ by nearly two days, facilitating complete grain-filling and enhancing the plant fruiting rate. The trend of $\bar{V}$ and V_max_ was similar, leading to more efficient decomposition of sucrose and starch synthesis in grains throughout the reproductive period. The EBR regulation of Tartary buckwheat maturation material allocation was outlined from several aspects: during the flowering stage, the CQ1 and XQ1 root share increased significantly, i.e., EBR promoted underground plant growth, and leaf dry matter weight proportion during the grouting period slightly increased. EBR also stimulated Tartary buckwheat flowering organ development, resulting in an increase in total flower number. The dry matter weight of seeds increased with EBR treatment, with a significant rise in seed proportion from the grain-filling period to maturity, followed by an equivalent increase in dry matter weight. From the percentage of dry matter weight, it can be observed that CQ1 emphasized allocating matter towards developing the root system and floral organs at anthesis, as well as to the leaves and seeds during the irrigating stage. On the other hand, XQ1 focused on developing stems and flowers during both anthesis and the irrigating stage, potentially contributing to the yield difference between the two. The plant “sources” and “reservoirs” interacted in a mutually beneficial and restrictive manner—optimal substance acquisition occurs when sources are ample and reservoirs are large. This highlights the importance of having sufficient sources and a large reservoir capacity for maximizing substance acquisition.

SPS utilized fructose-6-phosphate as a substrate in the functional leaves to convert it into sucrose phosphate, which was then transported to the grain for breakdown and starch synthesis [31]. EBR influenced the activity of key enzymes by modulating gene expression related to sucrose–starch metabolism, thereby controlling the synthesis, breakdown, and movement of sucrose–starch within the plant organs [32]. The results indicated that during the pre-grain-filling period (10–20 days), the treatment group exhibited higher sucrose levels compared to those of the control group, with increased SPS enzyme activity and enhanced sucrose synthesis in the Tartary buckwheat leaves. Additionally, the sucrose synthesis rate surpassed the translocation rate during this period. In the late grain-filling period (20–30 days), the sucrose content decreased in the treatment group relative to that of the control, accompanied by reduced SPS enzyme activity. Here, the rate of sucrose transport exceeded the rate of synthesis in the leaves, with EBR elevating sucrose transport in the Tartary buckwheat leaves during late grain-filling. Overall, during grain development, sucrose content and AI activity exhibited a negative correlation. Sucrose synthase (SS) is involved in the synthesis of sucrose, while acid invertase (AI) catalyzes the hydrolysis of sucrose into hexoses. In the early grain-filling stage (10–20 days post-anthesis), SS activity in the grain increased, leading to an elevated sucrose content, while AI activity diminished. At this point, free fructose and glucose from the leaves were converted to sucrose in the seed, with the rate of sucrose synthesis surpassing that of decomposition, leading to sucrose accumulation in the grain. During the middle and late stages of filling (after 20 days post-anthesis), SS activity continued to increase, along with AI activity. At this point, the rate of sucrose breakdown exceeded the rate of sucrose production, leading to a rise in starch synthesis. Consequently, the sucrose content in the seed grain-filling decreased significantly, while the starch content increased. The application of the EBR treatment enhanced the sucrose and starch synthesis rates in Tartary buckwheat seeds during both the pre-grain-filling and late-grain-filling stages by modulating SS and AI enzyme activity. This resulted in the increased storage strength and efficiency of the seed.

### 3.2. Effects of EBR on Photosynthetic Physiology of Tartary Buckwheat

Photosynthesis in the source leaves was crucial for producing carbohydrates to support growth [33]. The amount of photosynthetic pigments and leaf area directly impacted a plant’s photosynthetic capacity [34]. Prior research had demonstrated that EBR can regulate plant photosynthetic processes, enhancing assimilate production to meet growth requirements [35]. Photosynthesis relied on photosynthetic pigments for light absorption, electron transfer, and chemical reactions [36]. Previous studies have shown that EBR can improve plant photosynthesis. This is because EBR affects chlorophyll synthesis and photosynthesis and protects protein–pigment complexes, thereby reducing chlorophyll degradation [37,38]. This study found that EBR application increased chlorophyll content in Tartary buckwheat leaves during the grain-filling period, with optimal enhancement at the G2 treatment level. EBR treatment significantly boosted the net photosynthetic rate, stomatal conductance, and leaf transpiration in Tartary buckwheat, enhancing plant metabolism. The intercellular CO_2_ concentration initially decreased and then rose over time, with EBR promoting the photosynthetic rate and CO_2_ utilization, while reducing CO_2_ concentration. EBR can enhance the actual photosynthetic capacity of Tartary buckwheat during the grain-filling period, thereby enhancing light utilization efficiency and promoting photosynthesis.

### 3.3. Effect of EBR on the Content of Source Hormones in Tartary Buckwheat Grain-Filling Period

Phytohormones play vital roles in plant growth and development, influencing processes like cell division, tissue differentiation, flowering, and nutrient distribution. Plant growth regulators, including IAA, GA, and ABA, interact within complex signaling networks to coordinate responses to environmental and developmental cues. IAA is an important factor affecting the accumulation of dry matter in the grain. Higher IAA levels are beneficial to the differentiation of endosperm cells and the transfer of dry matter to the grain [39]. GA has a direct impact on the enrichment of seeds and the control of starch synthesis. A high GA content is beneficial to the dry matter accumulation of grains [40]. ABA promotes the conversion of sucrose into starch in the grain and shortens the grain-filling time, thus facilitating grain maturation [41]. The results of our study showed varying trends for ABA and GA_3_ concentrations when EBR was applied to Tartary buckwheat leaves at flowering onset. Notably, the ABA levels fluctuated in CQ1 and reduced in XQ1, while GA_3_ showed distinct trends in both regions. The treatments led to differences in seed retention post-maturity, possibly reducing grain loss. Additionally, the GA_3_ levels correlated with the EBR concentration, affecting flower numbers and enhancing photosynthesis in gibberellin-regulated plants. 

## 4. Materials and Methods

### 4.1. Experimental Design

The pot experiment was conducted in the greenhouse at Chengdu University from 2018 to 2019. The test material used was the conventional Tartary buckwheat cultivar Chuanqiao No. 1 (CQ1), which has a growth period of 86 to 93 days in Chengdu. This cultivar was bred by the Key Laboratory of Grains Processing under the Ministry of Agriculture and Rural Affairs at Chengdu University, Sichuan Province. The plant growth regulators tested included gibberellin (GA_3_), ethephon (ETH), Bijiu (B9), and 24-epibrassinolide (EBR), all of which were purchased from Shanghai Yuanye Biological Co., Ltd. (Shanghai, China). (Appendix A).

In this study, a completely randomized experimental design was employed. Trenches were dug vertically in the test field, measuring approximately 30 cm in width and one-third the height of the pots. The pots used were 25 cm in diameter and 20 cm in height. The test pots were then positioned in the ditches to replicate field conditions. Special soil was prepared, consisting of a volume ratio of natural soil to nutrient soil of 3:1, with each pot filled with 5 kg of this soil mixture. A total of 30 pots were arranged in each plot, organized into five rows of 6 pots each, and this setup was repeated three times, resulting in a total of 12 plots. Each pot was sown with 8–10 seeds, and after germination, the number of seedlings was thinned to three plants per pot at the five-leaf stage. During the initial flowering stage of Tartary buckwheat, the plants were treated with GA3 (25 mg/L), EBR (0.1 mg/L), B9 (2500 mg/L), and ETH (250 mg/L), while water served as the control (CK). The application of these treatments occurred between 16:00 and 18:00 in the evening, using a hand-operated sprayer to ensure thorough coverage of the entire plot. The Tartary buckwheat leaves were sprayed until the solution no longer dripped. Harvesting of the Tartary buckwheat seeds was conducted when 70% of the plants in each plot reached maturity.

Based on the pot experiments, effective EBRs were selected and subsequently tested in field trials at the Buckwheat Experimental Base of Chengdu University, located in Wufeng Town, Jintang County, Chengdu City, Sichuan Province, China (30°36′ N, 104°29′ E, elevation 411.5 m above sea level). The soil at the test site is classified as sandy loam, with a pH of 6.59. The nutrient composition of the 0–20 cm soil layer includes organic matter, 2.22%; total nitrogen, 1.29 g/kg; alkali hydrolyzed nitrogen, 128.0 mg/kg; total phosphorus, 0.87 g/kg; total potassium, 14.80 g/kg; available phosphorus, 23.7 mg/kg; and available potassium, 116.0 mg/kg. A completely randomized block design was employed to establish four concentration levels of EBR (selected from the pot experiments), i.e., 0.05 mg/L (G1), 0.1 mg/L (G2), and 0.2 mg/L (G3), with water serving as the control (G0). Before sowing, base fertilizers were applied, including nitrogen fertilizer at 45 kg/hm^2^, phosphate fertilizer at 120 kg/hm^2^, and potassium fertilizer at 40 kg/hm^2^. Nitrogen fertilizer was top-dressed at 45 kg/hm^2^ during the seedling stage. Each plot covered an area of 10 m^2^ (2 m × 5 m), with a 0.5 m interval between plots, and the experiment was replicated three times. Routine fertilizer and water management practices and daily monitoring for weeds, diseases, and insect pests were implemented. The average dosage of EBR spraying in the field was 10 m^2^ per 4 square meters.

### 4.2. Plant Sampling and Measurements

#### 4.2.1. Determination of Agronomic Traits and Yield Components

During the harvest period, data on basic agronomic traits were collected. From each plot, five plants exhibiting consistent growth under the same environmental conditions were selected. The measurements taken included plant height, the number of stem nodes, and the number of main stem branches. At the maturity stage, key yield-related traits of Tartary buckwheat were assessed, including the number of grains per plant, yield per plant, thousand-grain weight, and estimated yield per hectare.

Grain number per plant: Five plants were selected from each plot, and the number of grains per plant was counted. The average value was subsequently calculated.

Grain weight per plant: All mature seeds collected from these individual plants were dried in an oven at 60 °C until a constant weight was achieved, after which they were weighed.

1000-grain weight: The thousand-grain weight was determined by dividing the total yield per plant by the number of grains and multiplying this by 1000. The term “number of grains per plant” denoted the total count of mature grains produced by each plant.

Yield: For each plot, a 4-square-meter area of Tartary buckwheat was harvested and yield testing, and the yield per hectare was subsequently calculated.

#### 4.2.2. Flowering and Seed-Setting Rate Statistics

Flowering and seed-setting statistics: Five plants with uniform growth under the same environmental conditions were selected from each plot and the seed-setting rate, grain formation rate, and grain abortion rate were measured.
Seed-setting rate = number of mature grains/total number of flowers × 100%.(1)
Grain formation rate = (number of aborted grains + number of green grains + number of mature grains)/total number of flowers × 100%.(2)
Grain abortion rate = (number of aborted grains + number of green grains)/(number of aborted grains + number of green grains + number of mature grains) × 100%.(3)

#### 4.2.3. Grain-Filling Dynamics

The determination of the fitted growth equation for grain-filling in Tartary buckwheat involved sampling five plants with uniform growth from each plot every 7 days from the beginning of flowering until harvest. These plants were selected for their representativeness in determining the 100-grain weight of the seeds, ensuring an error margin of no more than 0.5 g. The Richards equation, which modeled Tartary buckwheat seed growth, i.e., W = A/(1 + Be^−Kt^)1/N, was used for fitting measurements. In this equation, A represents the maximum seed growth, B is the initial parameter value, K is the growth rate constant, and N is the curve bending parameter. The first-order derivative of the equation yields G = (KW)/N) × (1 − (W/A)^N^), allowing for the calculation of the starting grain-filling rate G0 when T is 0. This rate is used to determine the starting potential energy RG, where RG = G0/W0. Further calculations involve the second-order derivative to find the peak time T_max_ and the maximum grain-filling rate G_max_ [42].

Grain-filling weight changes: During the Tartary buckwheat flowering period, samples were taken every 10 days. Each sampling session involved selecting 10 plants with similar growth patterns to represent the plant population. The 100-grain weight of the seeds was determined for each sample, ensuring that the margin of error for the 100-grain weight did not exceed 0.5 g. This sampling process was repeated three times.

#### 4.2.4. Photosynthetic Parameters

The study assessed the impact of Tartary buckwheat photosynthetic pigment content at 10, 20, and 30 days post-anthesis. Samples were randomly collected from each plot, selecting five plants with uniform growth, free of pests and diseases. Functional leaves were extracted using the 80% acetone extraction method to measure chlorophyll a, chlorophyll b, and total chlorophyll concentrations [43].

The photosynthetic gas exchange parameters were determined using the GFS-3000 photosynthesis tester from Walz, Germany. Measurements were taken at 9:00–11:00 am on sunny days, selecting five plants with similar growth and environmental conditions.

#### 4.2.5. Dry Matter Mass

In Tartary buckwheat, six uniformly growing plants, free from pests and diseases, were randomly selected from each plot at 10, 20, and 30 days post-anthesis. The plants were categorized into roots, stems, leaves, flowers, and seeds, and their fresh weights were recorded. Subsequently, they were placed in an oven at 105 °C for 15 min to halt enzymatic activity, followed by drying at 60 °C until a constant weight was achieved. Finally, the dry weights were recorded.

#### 4.2.6. Starch–Sucrose-Related Synthase

Leaf sucrose– and seed sucrose–starch content were analyzed following the method outlined by Yue [44]. The activities of sucrose phosphate synthase, sucrose synthase, and acid converting enzyme were measured using Solarbio (Beijing Solarbio Science & Technology Co., Ltd., Beijing, China) kits. The functional leaves of Tartary buckwheat were sampled at 10 d, 20 d, and 30 d post-anthesis stages to assess the sucrose content and sucrose phosphate synthase (SPS) activity. Similarly, the sucrose and starch levels, along with sucrose synthase (SS) and acid invertase (AI) activities, were quantified in seeds at the same post-anthesis time points, each with three replicates.

#### 4.2.7. Determination of Endogenous Hormones in Seeds

Tartary buckwheat seeds were collected during the 20-day post-anthesis period for analysis. Endogenous hormones such as IAA, ABA, TZ, and GA_3_ were extracted from the seed samples using isopropanol-hydrochloric acid, while BL was extracted using methanol. Quantitative analysis of IAA, ABA, TZ, GA_3_, and BL in the seed samples was performed using the ESI-HPLC-MS/MS method. The analysis was carried out with an Agilent 1290 high-performance liquid chromatograph coupled with an AB Sciex QTRAP 650+ mass spectrometer. Internal standard substances were added during the extraction process to ensure the accuracy of the results.

### 4.3. Statistical Data

The raw data were analyzed using Excel 2022 (Microsoft Corp, Montgomery, AL, USA). LSD analysis and analysis of variance were conducted using SPSS 24.0 (IBM Corp, Chicago, IL, USA). Duncan’s multiple range test was employed to assess significant differences among samples (*p* ≤ 0.05). Graphs were generated using GraphPad Prism 9.4 (GraphPad Software Corp, San Diego, CA, USA). The R package generated the correlation network diagram in RStudio (version 4.4.1). Principal component analysis (PCA) of Tartary buckwheat-related indicators was conducted using Origin software (2024b).

## 5. Conclusions

As can be seen in Figure 7, in this study, Tartary buckwheat was treated with EBR at the onset of flowering to enhance leaf biosynthesis through increased photosynthetic pigment content, sucrose and SPS levels, and overall photosynthesis. The optimal concentration found to be effective was 0.1 mg/L, leading to a decrease in TZ, GA_3_, and ABA levels, while increasing IAA content. This resulted in higher starch, sucrose, SS, and AI content in the seeds, ultimately leading to increased grain weight and number per plant, thus enhancing overall yield. In future research, we can further investigate the molecular mechanisms by which EBR regulates gene expression during the grain-filling of Tartary buckwheat. Additionally, we can explore the chemical regulation of endogenous hormone levels in Tartary buckwheat through interactions with various exogenous hormones, aiming to identify methods to enhance current practices.

## Figures and Tables

**Figure 1 plants-13-03336-f001:**
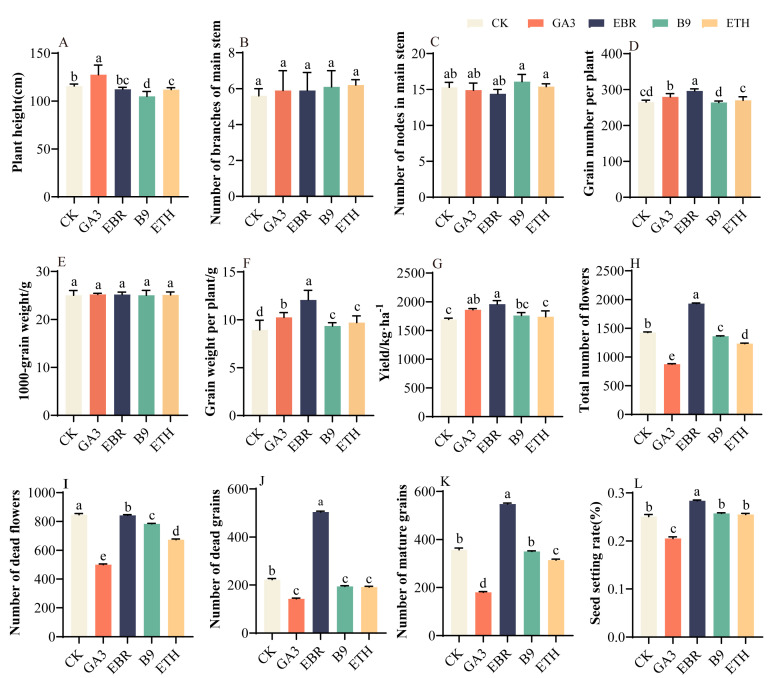
The agronomic traits (**A**–**G**) and seed-setting (**H**–**L**) of Tartary buckwheat under various growth regulator treatments. The parameters assessed include plant height (**A**), the number of main stem branches (**B**), the number of central stem nodes (**C**), the number of grains per plant (**D**), thousand-grain weight (**E**), yield (**G**), the total number of flowers (**H**), the number of dead flowers (**I**), the number of dead grains (**J**), the number of mature grains (**K**), and the seed-setting rate (**L**). Values in the graph reflect means; error bars show SEM. Different letters denote significant differences among means (*p* < 0.05).

**Figure 2 plants-13-03336-f002:**
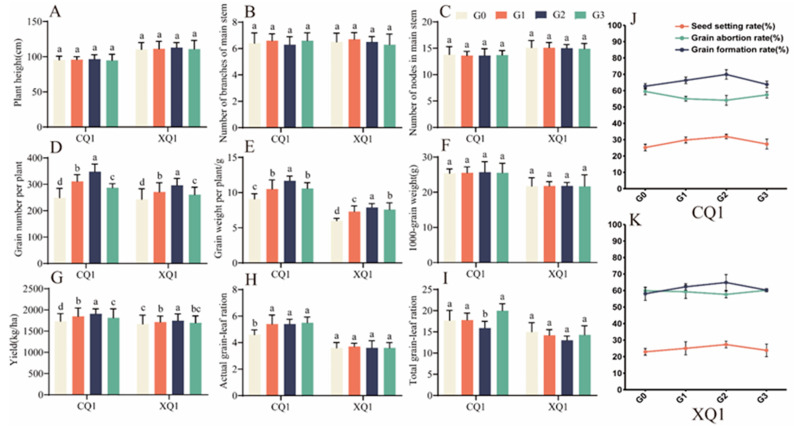
The agronomic traits (**A**–**I**), plant height (**A**), number of branches of main stem bran (**B**), number of nodes in main stem (**C**), grain number per plant (**D**), grain weight per plant (**E**), 1000-grain weight (**F**), yield (**G**), actual grain-leaf ratio (**H**), total grain-leaf ratio (**I**). as well as the fruiting conditions of CQ1 (**J**) and XQ1 (**K**), under varying concentrations of EBR treatments. CQ1 refers to Sichuan Buckwheat No. 1, while XQ1 denotes Xiqiao No. 1. G0, G1, G2, and G3 represent the four concentrations of EBR, i.e., 0 mg/L, 0.05 mg/L, 0.1 mg/L, and 0.2 mg/L. Values in the graph show the means; error bars show SEM. Different letters denote significant differences among means (*p* < 0.05).

**Figure 3 plants-13-03336-f003:**
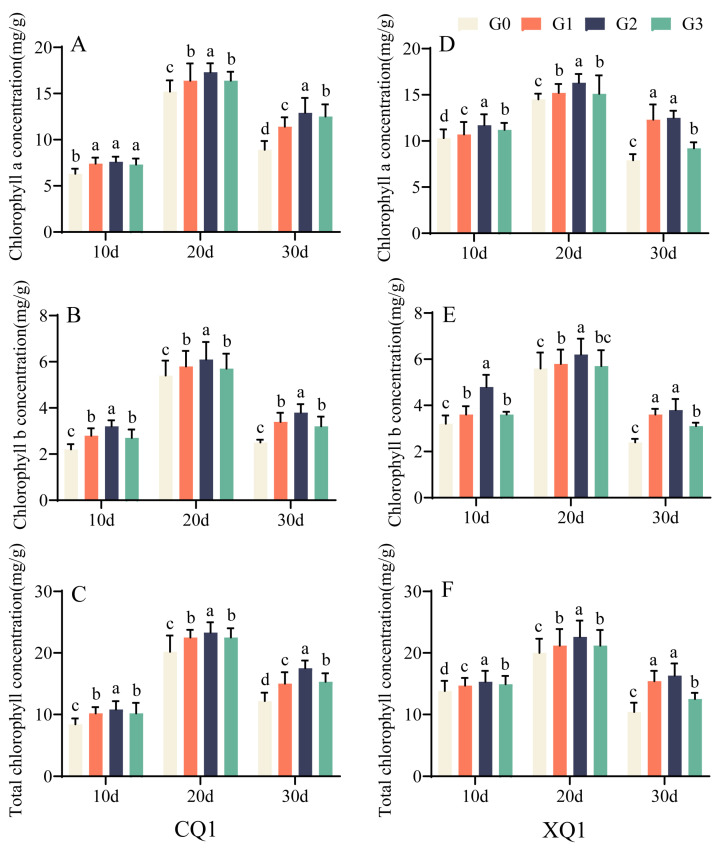
The chlorophyll content (AI) on various days under different EBR concentration treatments. The figure includes CQ1 chlorophyll a content (**A**), chlorophyll b content (**B**), and total chlorophyll content (**C**), as well as XQ1 chlorophyll a content (**D**), chlorophyll b content (**E**), and total chlorophyll content (**F**). Values in the graph reflect means; error bars indicate SEM. Different letters denote significant differences among means (*p* < 0.05).

**Figure 4 plants-13-03336-f004:**
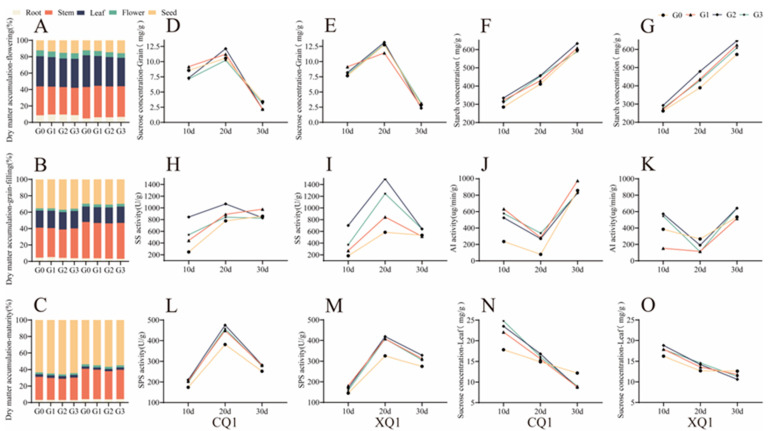
The enzyme content related to carbon metabolism (AL) under varying concentrations of EBR treatments. The accumulation of flowers, stems, and leaves are presented over three time points: 10 days (**A**), 20 days (**B**), and 30 days (**C**). Additionally, the grain sucrose content is shown for CQ1 (**D**) and XQ1 (**E**), followed by the grain starch content for CQ1 (**F**) and XQ1 (**G**). The figure further details sucrose synthase activity for CQ1 (**H**) and XQ1 (**I**), acid invertase activity for CQ1 (**J**) and XQ1 (**K**), and sucrose phosphate synthase activity for CQ1 (**L**) and XQ1 (**M**). Finally, the sucrose content in the leaves is represented for CQ1 (**N**) and XQ1 (**O**). CQ1 denotes Chuanqiao Buckwheat No. 1, while XQ1 refers to Xiqiao No. 1. Values in the graph reflect means (*p* < 0.05).

**Figure 5 plants-13-03336-f005:**
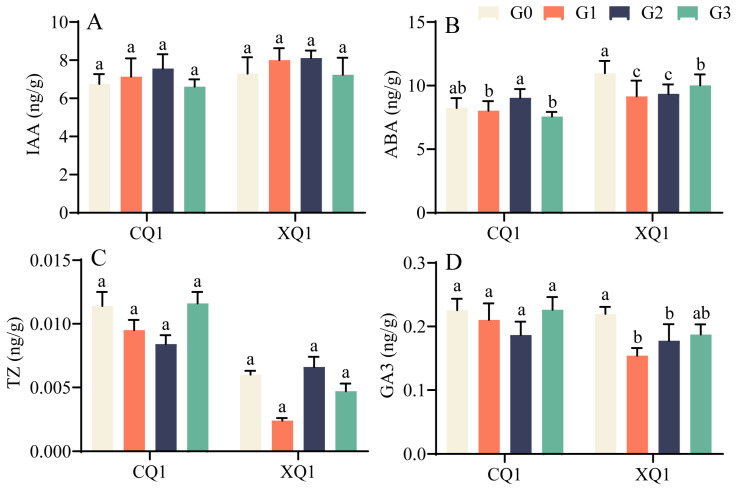
Chlorophyll concentration (**A**–**D**) on different days under different EBR concentration treatments. Indole-3-acetic acid (IAA) content (**A**); abscisic acid (ABA) content (**B**); zeatin (TZ) content (**C**); gibberellin (GA_3_) content (**D**). CQ1 stands for Chuanqiao No. 1; XQ1 stands for Xiqiao No. 1. Values in the graph reflect means; error bars indicate SEM. Different letters denote significant differences among means (*p* < 0.05).

**Figure 6 plants-13-03336-f006:**
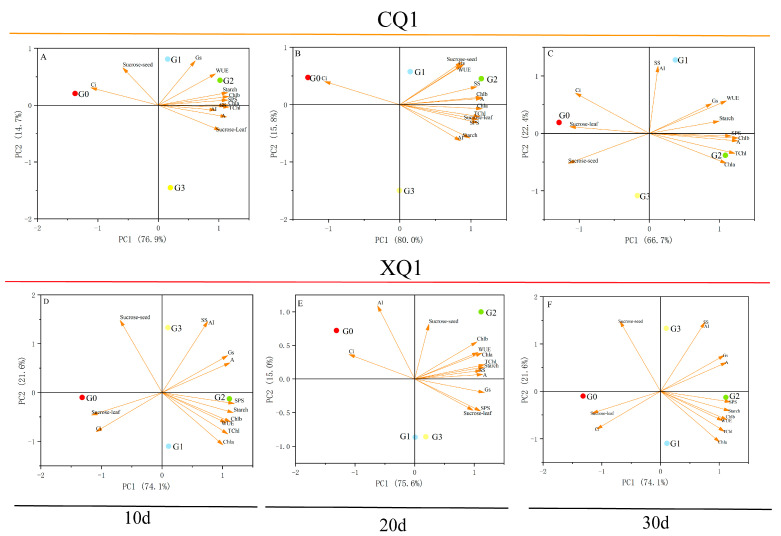
Biplot of CQ- and XQ1-related traits under different EBR concentration treatments. The indicators include Chla (chlorophyll a), Chlb (chlorophyll b), TChl (total chlorophyll), A (net photosynthetic rate), Gs (stomatal conductance), WUE (transpiration rate), Ci (intercellular CO_2_ concentration), SS (sucrose synthase), AI (acid invertase), starch (starch), SPS (sucrose phosphate synthase), sucrose-seed (grain sucrose), and sucrose-leaf (leaf sucrose).

**Figure 7 plants-13-03336-f007:**
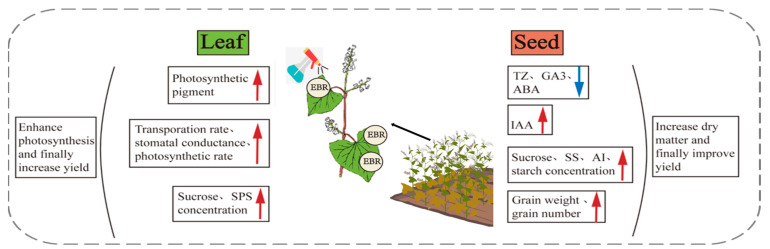
Mechanism diagram of EBR regulating yield increase of Tartary buckwheat. The red arrows indicate an increase, while the blue arrows indicate a decrease.

**Table 1 plants-13-03336-t001:** Parameters of the Richards equation for grain-filling of Tartary buckwheat after EBR treatment.

Variety	Treatment	A	B	K	N	R^2^	R0	V_max_ mg grain^−1^ d^−1^	T_max_ d	$\bar{V}$ mg grain^−1^ d^−1^
CQ1	G0	2.4259	0.1223	0.0542	0.0327	0.9963	7.622	0.1688	19.5221	0.1237
	G1	2.5124	0.1455	0.0795	0.0319	0.9981	9.216	0.1754	18.622	0.1366
	G2	2.5537	0.1573	0.0731	0.0388	0.9979	9.884	0.1921	17.5525	0.1503
	G3	2.4776	0.1384	0.0664	0.0374	0.9931	8.462	0.1886	19.3634	0.1441
XQ1	G0	1.7273	0.0594	0.0422	0.0264	0.9949	5.334	0.1267	22.1776	0.0775
	G1	1.8005	0.0751	0.0614	0.0255	0.9961	7.514	0.1421	21.9654	0.0834
	G2	1.8097	0.0866	0.0751	0.0292	0.9982	8.022	0.1613	20.7422	0.0902
	G3	1.7874	0.0672	0.0629	0.0248	0.9917	7.119	0.1485	22.0617	0.0867

A is the final biomass of grain-filling; B, K, and N are all basic parameters of the Richards equation; and R^2^ is the test judgment parameter. R0 is the starting potential energy of grain-filling, V_max_ is the maximum grain-filling rate, T_max_ is the time it takes for grains to reach the maximum grain-filling rate, and V is the average grain-filling rate (*p* < 0.05).

**Table 2 plants-13-03336-t002:** Effect of EBR on photosynthetic parameters of Tartary buckwheat.

Date	Variety	Treatment	A	Gs	WUE	Ci
10d	CQ1	G0	21.76 c	0.67 bc	7.60 b	503.1 a
		G1	23.32 b	0.68 ab	7.76 a	490.2 b
		G2	25.66 a	0.69 a	7.87 a	475.5 c
		G3	24.54 ab	0.66 c	7.62 b	479.5 c
	XQ1	G0	19.54 c	0.62 b	6.64 c	351.9 c
		G1	20.99 b	0.63 a	6.83 b	360.5 b
		G2	23.31 a	0.64 a	6.95 a	349.1 c
		G3	21.63 b	0.64 a	6.69 c	376.9 a
20d	CQ1	G0	24.91 c	0.52 b	6.29 b	397.5 a
		G1	28.61 b	0.54 a	6.44 a	386.6 b
		G2	30.99 a	0.54 a	6.48 a	367.8 c
		G3	27.69 b	0.52 b	6.28 b	375.3 c
	XQ1	G0	22.30 c	0.46 c	5.29 c	325.6 a
		G1	25.92 b	0.49 b	5.48 b	308.5 b
		G2	27.87 a	0.51 a	5.68 a	298.5 c
		G3	24.71 b	0.50 a	5.37 bc	298.4 c
30d	CQ1	G0	22.945 c	0.42 b	4.21 b	467.7 a
		G1	26.08 b	0.43 a	4.42 a	457.1 a
		G2	28.07 a	0.44 a	4.45 a	437.2 b
		G3	25.44 b	0.41 b	4.21 b	445.3 b
	XQ1	G0	18.59 c	0.35 c	3.25 c	395.4 a
		G1	21.36 b	0.38 b	3.38 b	379.7 b
		G2	23.06 a	0.40 a	3.56 a	366.3 c
		G3	22.72 a	0.40 a	3.29 bc	367.2 c

Date represents the number of days post-flowering. A (µmol m^−2^·s·^−1^) denotes the net photosynthetic rate, while Gs (mol·m^−2^·s·^−1^) indicates the stomatal conductance. WUE (mmol·m^−2^·s·^−1^) reflects the transpiration rate, and Ci (µmol·mol^−1^) signifies the intercellular CO_2_ concentration. Different lowercase letters in the same column and within each date indicate significant differences (*p* < 0.05).

## Data Availability

All datasets supporting the conclusions of this article are included within the article. If not included in the manuscript, they are available from the corresponding author upon reasonable request.

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
