# Peer review of "Physiological Mechanism of EBR for Grain-Filling and Yield Formation of Tartary Buckwheat"

_plants, 2024, doi:10.3390/plants13233336_

Round 1
Reviewer 1 Report
Comments and Suggestions for Authors
This paper reported the positive effects of and physiological mechanisms related to application of brassinosteroids (EBR) on grain filling of Tartary buckwheat. A preliminary pot experiment was set up to test the efficiency of various hormones, and then EBR was selected for the field experiment. The experimental data were interesting, and it was convincing that EBR at 0.1 mg/L was beneficial but there are many points that should be addressed more clearly.
1) The discussion, Lines 330 -340, dealing with changes in sucrose/ starch content in developing grains in relation to activity of SS and AI enzymes is confusing, please rewrite. Also, the sentence “At this point, free fructose and glucose from the leaves were converted to sucrose in the seed, with the rate of sucrose synthesis surpassing decomposition, leading to sucrose accumulation in the grain.” This sentence needs a reference(s), normally leaves do not export sugar in the form of free glucose and fructose?
2) Please also improve the discussion, Lines 366-376, on the effects of EBR on changes in endogenous hormones. In its current form, it only repeated the results without discussing the interaction among hormones and the effects of hormones on grain development/filling.
3) Figure 1G – please re-check the alphabet above the bar graphs. Figure 1L – the unit of seed setting rate should be %? Figure 1J – please discuss why EBR treatment profoundly enhanced the number of dead seeds.
4) Table 2 – why were the values of Fv/Fm at 10d much lower than 0.8 which is the optimum value for healthy leaves? And also, the values of Fv/Fm higher than 1.0 at 20d seems strange.
5) Figure 3 – the units of chlorophyll content should be mg/g leaf fresh weight not mg/L
6) Heading for 2.3.2… should be Effects on dry matter conversion and enzyme activity
7) Figure 5 – the legend was wrong. Please also discuss the dramatically high content of BL in Figure 5E.
8) Materials and methods – please provide the reference(s) for the method of calculation of grain-filling dynamics, and measurement of chlorophyll contents.
9) Materials and methods – Line 483 what is ‘maximum photochemical yield of PSII (FvII)’ ?
10) Abstract – ‘The results indicated that EBR treatment followed the typical 'S' type growth curve of crops, resulting in an increase in the Tartary buckwheat grain-filling rate.’ There is no mention about ‘S growth curve’ anywhere in the content?
11) Abstract – Lines 28-30- ‘Specifically, EBR increased the activity of the pre-leaf sucrose phosphate synthase (SPS) enzyme, seed sucrose synthase (SS) enzyme, late grain-filling acid invertase (AI) enzyme, grain-filling SPS enzyme, and SS enzyme’. Please improve this sentence to be more specific about which enzymes referred to leaves, which for developing seeds, and at which stage?
Author Response
Hello, thank you for your opinion. My reply is in word. Please see the attachment.

Reviewer 2 Report
Comments and Suggestions for Authors Dear Editor
The result content the information found interesting the effect of EBR, the others grow regulators have minor effect.
But the Ms need to be written with expert in writing Ms. I wrote.
CK,- clean water USED AS CONTROL. It’s a pure distilled water or not? Clarify.
The effect of four concentration of EBR: 0.05 mg/L (G1), 0.1 mg/L (G2), and 0.2 mg/L (G3), with clear water serving as the control (G0) were used for treatment.
In Fig 1.- The growth regulator EBR: increased the number of flowers, it induced more than 300 death of grains, it enhanced 170 mature grains approximately, all data compared to control clean water CK. The data for the others experiments EBR have no-effect.
In Fig. 2. The agronomic traits (AI), as well as the fruiting situations of CQ1 (J) and XQ1 135 (K), treated with various concentrations of EBR, the results show no statistical difference with the control, and therefore no effect of the growth regulator EBR is observed.
In the phrase:
“Under G2 treatment, the CQ1 and XQ1A values in-142 creased by 5.26% and 4.77%, respectively, compared to the control group. EBR enhances the initial grouting potential of CQ1, with the R0 values for XQ1 and CQ1 144 being highest under the G2 treatment.”
The approx. 5 % increases induced with EBR is low and with the concentration of G3 EBR decreased almost to control.
In the phrase:
“EBR treatment reduced Tmax, with the 149 most significant reduction observed at the G2 concentration; specifically, CQ1 and 150 XQ1 experienced reductions of 1.97 days and 1.44 days, respectively.”
The almost 1 day reduction in the grain filling induced with 0.1 mg/L G2 EBR is seems to be not important?
In Table 2 shows that 0.1 mg/L G2 EBR for 20 days promoted the net 165 photosynthetic rate of Tartary buckwheat during the grain-filling period, with the 166 effect being more pronounced in CQ1 and the treatment with 0.2 mg/L G3 EBR decreased net photosynthesis rate.
As illustrated in Figure 3A-F, the concentrations of chlorophyll a, 181 chlorophyll b, and total chlorophyll increased from 20 days post-flowering, lower chlorophyll content at 10 and 30 day growth. Thus, 183 0.1 mg/L EBR minor effect of photosynthetic capacity of Tar-184 tary buckwheat was found during grain filling.
Figure 4 A-C shows that during the flowering stage of Tartary buckwheat, the accumulation of flowers, stems, and leaves, during 10 days (A), 20 days (B), and 30 days (C) of grow. Flowers production increases as days of grow increases. However the leaves chlorophylls at 10 day decreased as time increases, being very low leaves chlorophyll concentrations at 30 day of grow. Notice that stem production at 10 day of grow decreased drastically over 30 days of grow.
When the flowering stage of Tartary buckwheat was treated with EBR shown that the grain sucrose content for CQ1 (D) and XQ1 (E) increases at 10 to 20 days of grow and at 30 days of grow decreased the sucrose content. Notice that when EBR concentration increased have no effect in sucrose content. Finally the grain starch content for CQ1 (F) and XQ1 (G) increased as days of grow increased. However when EBR concentration increased have no-effect on grain starch content.
Further studies in the sucrose phosphate synthase activity for CQ1 (L) and XQ1 (M) show that it activity increased at 20 days of grow and decreased at 30 days of grow. In the same figure show small increase in sucrose phosphate synthase activity with various concentrations of EBR.
Finally, Fig 4 N show that 240 sucrose content in leaves represented for CQ1 (N)
and XQ1 (O) the highest sucrose content at 10 days of grow and decreased linearly the sucrose content as days of grow increased. Also the treatment with different concentrations of EBR have no-effect.
In addition sucrose synthase activity for CQ1 (H) and XQ1 (I) increased the effect of EBR from 10 to 20 days of grow, thereafter decreased. Furthermore when EBR increased from 0.5 to 0.2 mg/L enhanced the activity. But in acid invertase activity for 239 CQ1 (J) and XQ1 (K) at 10 and 30 days of Tartary buckwheat grow were enhanced with different concentration of EBR, but at 20 days of Tartary buckwheat grow the activity decreased.
These phrase need to be incorporated with the results: “CQ1 202 allocates more resources to roots and flowers than XQ1, which directs more resources 203 to stems and leaves. Notably, CQ1 exhibits more leaves during the grain-filling stage 204 than in the flowering stage, indicating a shift in the center of gravity of material 205 transport toward the grain. The intensity of grain filling in CQ1 surpasses that of XQ1. 206 Conversely, during the grain-filling stage, XQ1 prioritizes material allocation to-207 wards developing flowers and stems over CQ1. Additionally, EBR treatment en-208 hances the reproductive growth and development of Tartary buckwheat during the 209 grain-filling period, significantly increasing the proportion of grains in both CQ1 and 210 XQ1 under G2 treatment. Compared to the control group, the treatment group 211 demonstrates an increase in the dry matter weight of the entire plant and promotes 212 a higher grain dry matter distribution ratio. 213
• As illustrated in Figure 4D-O, EBR treatment significantly enhances the chemical ac-214 tivity of SPS. The SPS activity in functional leaves during the grain-filling stage peaks 215 at 20 days after flowering, with G2 exhibiting the highest activity and EBR”
In Fig 5. In the graph TZ vc XQI EBR decreased the activity of TZ at concentration of 0.05 and 0.2 mg/L than control. In the same Fig 5 graph BL vc CQI at 0.2 mg/L EBR enhanced highly the activity of BL than control, in the same graph in XQI EBR at 0.05 and 0.2 mg/L inhibited the activity of BL than control.
Author Response

(The authors gave the same response as above.)

Reviewer 3 Report
Comments and Suggestions for Authors
In this study the effects of different hormones application on buck wheat plant growth as well as on important agronomic traits are reported.However the interesting data presented on the enhancement of the activity of key-enzymes related to sucrose-starch metabolism as well as on the accumulation of photosynthetic products during the grain- filling period associated to BR administration do not allow conclusion of a cause -effect relationship between the hormone and the observed physiological parameters.
In the legenda ofTable1 an2 I would specify the initials of the parameters present on the top of the Table
Author Response

(The authors gave the same response as above.)

Reviewer 4 Report
Comments and Suggestions for Authors
However, a review must be carried out by the authors for publication. The following are the suggested observations for approval:
Abstract: In one sentence, explain your study's general observation for conclusion.
Introduction: The introduction section needs major revision and clearly states the problems of the existing works. The author should show us a deep analysis of the gaps in existing methods and why your research is essential. The study's novelty should be highlighted, and the topic should be focused on in-depth. Deepen the role of EBR in plants.
Lines 64-68: analyze the need to cite other hormones
Review the introduction, because there are loose sentences with no connection between them.
In practice, how is EBR used? What studies have been done with this type of hormone in similar plants?
Line96: Check the differences in Figure 1 G- because EBR is bc and CK is c
Abbreviations must be specified in the titles of all Figures and Tables.
Page 9 - Figure 5
All abbreviations must be inserted in the chart title. Ex.: IAA, ABA...G0, 1, 2 3
Check if it is from A-L
Line 290: The discussion section should be improved by adding the most recent related studies with proper justification to support your results.
Page 13, line 397: Insert explanation of dosage selection
Line 419-...: The methodological information already included in the first experiment does not need to be repeated.
Line426: Plant sampling and measurements: For which experiment?
Line 429: What does mean: "under environmental conditions"?
Line470: The photos were not presented in the article, why?
Line 485: Specify?
Line 519: The conclusion can be improved. What are your recommendations and suggestions based on your analysis for further studies?
Line 520: The Figure is interesting, but it will need to be inserted in another location, I suggest in the discussion. Furthermore, the sentence refers to a field trial, which is not represented by the figure.
Author Response

(The authors gave the same response as above.)

Reviewer 5 Report
Comments and Suggestions for Authors
The manuscript studies the effects of brassinosteroids (EBR) on Tartary buckwheat varieties with differing seed-setting rates. It aimed to understand how EBR influences grain-filling, seed-setting rates, and the underlying physiological mechanisms. Results showed that EBR treatment increased the grain-filling rate, with the high seed-setting variety demonstrating superior capacity. The optimal concentration of EBR (0.1 mg/L) significantly enhanced seed-setting rates and yield by regulating hormone levels and improving photosynthesis. Respectfully, I recommend 7-50you the following aspects to improve the quality of the manuscript significantly:
- Ensuring that the introduction or initial background clearly describes the research objective and hypothesis and the importance of using EBR and other treatments. Current results move on to specific findings without smoothly introducing the broader objectives.
- some references, such as "Figure 1," are used at the beginning, but details about these figures (what exactly is illustrated) are missing. Provide more apparent comments or legends directly interpreting the figures and what they indicate about the treatments. Instead of stating that EBR increased various parameters, specify the biological relevance of these changes. -When describing increases or decreases (e.g., plant height, seed number), state whether these differences are statistically significant and provide p-values ​​or confidence intervals where appropriate.
-Ensure that principal component analysis (PCA) results are interpreted, explaining how the clustering or separation in the PCA plots relates to the overall analysis. Define key terms such as PC1 and PC2 in context, making it accessible to readers unfamiliar with PCA.
-Standardize plant variety abbreviations (CQ1, XQ1). Ensure that these abbreviations are used consistently and defined at the beginning of the text.
-You should provide consistent descriptions of treatments (G2, G3, CK) in a separate section or a clear legend. Explain acronyms at the beginning of the manuscript.
-The discussion of the results section should be improved, placing the findings in the context of the existing literature. How do the effects of EBR on Tartary buckwheat compare to other crops or similar studies?
- Please provide further explanations on the mechanisms through which EBR influences photosynthesis, starch accumulation, and hormone levels, which are well supported by the literature. Can any information be extracted on how EBR affects the expression of genes related to these pathways?
Finally, I found some sentences coinciding with other literature references: Lines 27-29, 47-50, 435-437, 487-492, 512-515. Please modify these sections of the manuscript.
Comments on the Quality of English Language
- some sentences need more fluid transitions, making it easier to follow the progression of ideas. Break complex results into more digestible sections with better flow.
-Repeated phrases are evident. Avoid repeatedly starting sentences with phrases such as "tartar buckwheat" or "EBR treatment." Varied sentence structure improves readability.
Author Response

(The authors gave the same response as above.)

Round 2
Reviewer 3 Report
Comments and Suggestions for Authors
Some minor considerations:
line 78"However, involved in many physiological..."This sentence should be reformulated
line 95:delete"is"
line45: gor?
lie 115: Is this statement correct? can òyoy reformulate
line 160: grouting ?
line 337:" the presence of exogenous ...."
sounds a speculation
line 312 " :qu ote; S quote: ?
388: "played" should be changed to "play"
line 397 "interacted" should be changed to " interact"
Comments on the Quality of English LanguageEnglish can be improved ,particularly so in the DISCUSSION
Author Response
Dear reviewers, thank you for your careful review and constructivesuggestions regarding our manuscript. We have revised the manuscript inaccordance with the comments and marked all the amends on our revisedmanuscript.

Reviewer 5 Report
Comments and Suggestions for Authors
The authors improved the last version of the manuscript, giving satisfactory answers to my comments.
Author Response
Dear Editors, Thank you for your kind letter and your careful work regarding our manuscript.
